# Exercise for people living with frailty and receiving haemodialysis: a mixed-methods randomised controlled feasibility study

Hannah M L Young ,[1,2] Daniel S March,[3] Patrick J Highton,[3,4] Matthew P M Graham-Brown,[3,4] Darren C Churchward,[3] Charlotte Grantham,[3] Samantha Goodliffe,[5] William Jones,[6] Mei-Mei Cheung,[7] Sharlene A Greenwood,[8,9] Helen C Eborall,[10] Simon Conroy,[5] Sally J Singh,[11,12] Alice C Smith,[5] James O Burton[3,4]

► Prepublication history and additional materials for this paper is available online. To view these files, please visit the journal online (http://dx.doi.org/10.1136/bmjopen-2020-041227).

For numbered affiliations see end of article.

**Correspondence to**
Hannah M L Young;
hmly1@le.ac.uk

### ABSTRACT

**Objectives** Frailty is highly prevalent in haemodialysis (HD) patients, leading to poor outcomes. This study aimed to determine whether a randomised controlled trial (RCT) of intradialytic exercise is feasible for frail HD patients, and explore how the intervention may be tailored to their needs.

**Design** Mixed-methods feasibility.

**Setting and participants** Prevalent adult HD patients of the *CYCLE-HD* trial with a Clinical Frailty Scale Score of 4–7 (vulnerable to severely frail) were eligible for the feasibility study.

**Interventions** Participants in the exercise group undertook 6 months of three times per week, progressive, moderate intensity intradialytic cycling (IDC).

**Outcomes** Primary outcomes were related to feasibility. Secondary outcomes were falls incidence measured from baseline to 1 year following intervention completion, and exercise capacity, physical function, physical activity and patient-reported outcomes measured at baseline and 6 months. Acceptability of trial procedures and the intervention were explored via diaries and interviews with n=25 frail HD patients who both participated in (n=13, 52%), and declined (n=12, 48%), the trial.

**Results** 124 (30%) patients were eligible, and of these 64 (52%) consented with 51 (80%) subsequently completing a baseline assessment. n=24 (71% male; 59±13 years) dialysed during shifts randomly assigned to exercise and n=27 (81% male; 65±11 years) shifts assigned to usual care. n=6 (12%) were lost to follow-up. The exercise group completed 74% of sessions. 27%–89% of secondary outcome data were missing. Frail HD patients outlined several ways to enhance trial procedures. Maintaining ability to undertake activities of daily living and social participation were outcomes of primary importance. Participants desired a varied exercise programme.

**Conclusions** A definitive RCT is feasible, however a comprehensive exercise programme may be more efficacious than IDC in this population.

**Trial registration numbers** ISRCTN11299707; ISRCTN12840463.

### Strengths and limitations of this study

► To the best of our knowledge, this is the first study to evaluate the feasibility of an exercise intervention for people living with frailty and receiving haemodialysis (HD).

► The Clinical Frailty Scale, a frailty risk-stratification measure which has been validated in an HD population, was used to identify eligible participants.

► This study is also the first to explore how trial procedures and exercise programmes should be specifically tailored to the needs of people living with frailty and receiving HD, from their own perspectives.

► Multiple qualitative methods (interviews and diaries) were used to explore participants perceptions, providing a form of triangulation which strengthens the conclusions made.

► Due to the nature of the intervention and resource limitations, we could not blind intervention providers, outcome assessors or study participants to group allocation.

### INTRODUCTION

Frailty, 'a multidimensional syndrome of decreased physiological reserve leading to increased vulnerability to minor health stressors', is highly prevalent within the haemodialysis (HD) population.[1 2] Increasing frailty is associated with worsening outcomes, including mortality, hospitalisation, falls, reduced Health-Related Quality of Life, psychological well-being, physical function, ability to undertake activities of daily living (ADLs) and increased symptom burden.[3–5]

Despite this, frailty is not static and evidence suggests that some factors associated with frailty are amenable to change.[6] While the possible mediating role of exercise has been discussed, to the best of our knowledge, no

original studies have examined the feasibility or effectiveness of an exercise programme for people living with frailty and receiving HD.[7] To date, exercise interventions for HD patients have focused on intradialytic exercise, most commonly delivered by means of a cycle ergometer (intradialytic cycling, IDC), yet little is known about whether this is the most appropriate training stimulus for frail HD patients.[8] In addition, HD treatment can be poorly tolerated by frail patients and therefore IDC may represent an additional stressor to which these patients are particularly vulnerable.[9] European renal best practice guidance highlights a need for studies which identify how exercise programmes should be more specifically tailored to the needs of frail patients with chronic kidney disease (CKD),[10] yet to date, there has also been no exploration of the needs, barriers and facilitators to exercise from the perspectives of people living with frailty and receiving HD themselves.

The aim of this study was to determine the feasibility of conducting a randomised controlled trial (RCT) investigating the effects of IDC for HD patients living with frailty by: (1) estimating rates of eligibility, recruitment, retention, exercise adherence and outcome acceptability; and exploring (2) the potential benefits of IDC across a range of secondary outcomes and (3) the perceptions of frail HD patients in relation to participating in clinical research, IDC and a tailored exercise intervention.

## METHODS

### Design

A prospective, randomised controlled feasibility study was conducted alongside concurrent qualitative diaries and interviews. The feasibility study was a secondary analysis of the *CYCLE-HD* trial, whose aims and methods are reported elsewhere.[11] The qualitative component was underpinned by a constructivist Grounded Theory approach.[12] All participants provided written informed consent.

### Participants

Prevalent adult (over 18 years) HD patients were recruited from three centres within the UK East Midlands Renal Network. In addition to the inclusion and exclusion criteria for the *CYCLE-HD* trial (see online supplemental material 1), the Clinical Frailty Scale (CFS), a risk stratification tool, was used to identify vulnerable to severely frail participants (CFS score 4–7).[13] The CFS has good predictive abilities in an HD population, good construct validity when compared with the Frailty Index, is less burdensome that the Frailty Phenotype and has been validated in an HD population.[13–15]

The inclusion and exclusion criteria for the qualitative component mirrored the feasibility study and both those involved in the trial, and those who were eligible but declined to participate, were eligible.

### Randomisation

HD cohorts were randomised prior to screening, based on a computer-generated randomisation algorithm held by the Robertson Centre for Biostatistics at the University of Glasgow.

### Recruitment

Patients were screened for eligibility by their supervising nephrologist. Suitable patients were approached during HD, and the study explained. For the qualitative component, participants who had been involved in the feasibility study were recruited following completion of, or withdrawal from, the trial to prevent contamination.

### Exercise intervention

Online supplemental material 2 outlines the exercise intervention in line with TIDieR guidance.[16] Briefly, following a 1-month run-in, participants in the exercise group undertook three times per week supervised, moderate-intensity (Rating of Perceived Exertion, RPE 12–14) IDC (MOTOmed Letto2, Reck, Germany), for 6 months.[17] Cycling resistance was progressively increased to maintain RPE in response to exercise adaptation. Both arms continued with usual care HD as described elsewhere.[11]

### Sample size

Determinations of sample size from a power calculation around a primary outcome are not relevant to a feasibility study and sample sizes of 24–50 are considered sufficient.[18] For the qualitative component, maximum variation sampling was initially used to ensure diversity in frailty status and level of trial participation.[12] As understanding was gained from preliminary analyses, theoretical sampling was used to further recruit participants.[12] A maximum of 30 interviews were planned, but data collection ceased at the point where theoretical categories were saturated and no longer generated new insight (n=25).

### Primary outcome measures

The primary feasibility outcomes are presented in online supplemental material 3. Judgement regarding feasibility was based on a set of *a priori* progression criteria. For each criterion, the development of 'stop' (indicating when there are issues with the trial that cannot be resolved) and 'go' thresholds (when there are no issues that may impede the success of a trial) were co-produced by patients, clinicians and researchers.[19 20] Results falling between these thresholds indicated that adaptation to trial procedures may render a definitive RCT viable.[20]

### Baseline demographic and clinical variables

Demographic and clinical characteristics were gathered from participants' medical notes. The Charlson Comorbidity Index was used to estimate the burden of comorbid disease.[21]

### Secondary outcome measures

Multiple secondary outcomes were used to determine the potential effects of IDC and most appropriate primary endpoint for a future RCT. Outcome assessors were not blinded to group allocation.

Information on the number of falls, defined as 'an unexpected event in which the participants come to rest on the ground, floor, or lower-level' which resulted in Emergency Department visits and hospital admissions were collected from baseline to 1 year following intervention completion from medical records and hospital episode statistics.[22]

Field tests of exercise capacity and physical function included the Incremental Shuttle Walk Test (ISWT), the Endurance Shuttle Walk Test (ESWT), the Short Physical Performance Battery (SPPB) and the Sit-to Stand in Sixty Seconds (STS60).[11] Physical activity (PA) was objectively measured using the SenseWear Armband Pro 3 (BodyMedia, Pittsburgh PA, USA) for seven consecutive days, including HD. Established criteria were used to ensure representative data for average daily wear-time, steps per day and time (minutes per day) spent in sedentary (defined as 0–1.5 metabolic equivalents (METS)), light (1.6–2.9 METS) moderate (3–6 METS) and vigorous (>6 METS) PA.[23] Patient-reported outcomes (PROMs) collected are outlined in online supplemental material 4.[11] All outcomes were collected at baseline and 6 months.

Serious adverse events (SAEs) were recorded and assessed from baseline to 6 months as outlined previously.[11]

### Diaries and interviews

Participants first completed a prospective falls diary, recognised as the current 'gold standard' for falls data collection, for up to 3 months to examine the feasibility of this outcome measure within a future definitive RCT.[22] Semistructured interviews then explored participants' experiences of: (1) keeping a falls diary; (2) participating in a trial and (3) their perceptions of IDC and a tailored exercise intervention.

Information to support diary collection and a topic guide for the interviews (see online supplemental material 5) was developed by HY, HCE and a patient and public involvement (PPI) group. Topics were tailored according to the level of involvement in the trial, and the content of diaries. Interviews were conducted during HD, in the participant's home, or in the hospital by HY and lasted 20–120 min (mean 63 min). All were digitally audio-recorded and transcribed verbatim.

### Data analysis

Sample characteristics are presented as mean±SD, median (IQR) or n (%), as appropriate. Descriptive statistics and confidence intervals were used to estimate feasibility outcomes.[24] The percentage of exercise sessions completed was used to establish the acceptability of IDC. Outcome acceptability was determined by quantifying the amount of missing data across secondary outcomes. No imputation was performed to account for missing data. No statistical testing relating to the efficacy of the exercise intervention was undertaken, although the potential benefits of exercise were estimated.[24] For falls, summary data, IRR (the ratio of the incidence rate in the exercise group divided by the incidence rate in the usual care group) and 95% CIs were presented. Statistical analyses were performed using SPSS V.24 and Stata V.16.

Qualitative analysis was undertaken by HY and SG and informed by a constant comparative approach.[12] Transcripts were reviewed, then coded line by line, followed by focused and then theoretical, coding.[12] NVivo V.11 software (QSR International Ltd, version 11, 2016) was used to facilitate data management. Finally, qualitative and quantitative results were merged in a 'joint display' to facilitate an overall assessment of feasibility.[25]

### Patient and public involvement

The PPI group for this study comprised patients of all ages, genders and ethnicities who were living with frailty and receiving HD, and their relatives. They agreed this study was an important priority for further investigation and particularly stressed the need to add the qualitative component. The PPI group were involved early in the ethical approval stages and were actively engaged in writing lay summaries and providing patient perspectives on data collection procedures, ethical issues and the study dissemination plans. They assisted in the preparation of study documentation, interview topic guides and diary keeping materials. During the study, members of the PPI group attended regular steering meetings and were involved in co-producing the progression criteria.

## RESULTS

### Feasibility study

#### Eligibility and recruitment

Screening and recruitment took place from March 2015 to 2018, with data collection completed by November 2018. Figure 1 outlines the trial Consolidated Standards of Reporting Trials. Of the 406 patients screened in the *CYCLE-HD* trial, n=124 (30%, 95% CI 26.1% to 35.3%) were identified as vulnerable to severely frail and therefore eligible for the feasibility study. Sixty-four participants (52%, 95% CI 42.5% to 60.7%) consented. Reasons for declining were lack of time or family support and reluctance to undergo outcome testing, or to be randomised. Those who declined to participate had a median age of 73 (IQR 67–81) years. N=35 (58%) were female and n=25 (42%) male. Twenty-five (42%) were classified as vulnerable according to the CFS, n=17 (28%) were mildly frail, n=9 (15%) moderately frail and n=9 (15%) severely frail. Thirteen (20%, 95% CI 11.3% to 32.2%) participants withdrew prior to baseline assessment. N=51 (80%, 95% CI 67.8% to 88.7%) completed this assessment. Twenty-four (47%) participants received dialysis during shifts randomised to exercise and 27 (53%) during shifts randomised to usual care.

#### Participant characteristics

Table 1 displays the characteristics of the trial participants at baseline. Groups were well matched across most variables. A lower proportion of participants were female (23.5%) and severely frail (6%) overall.

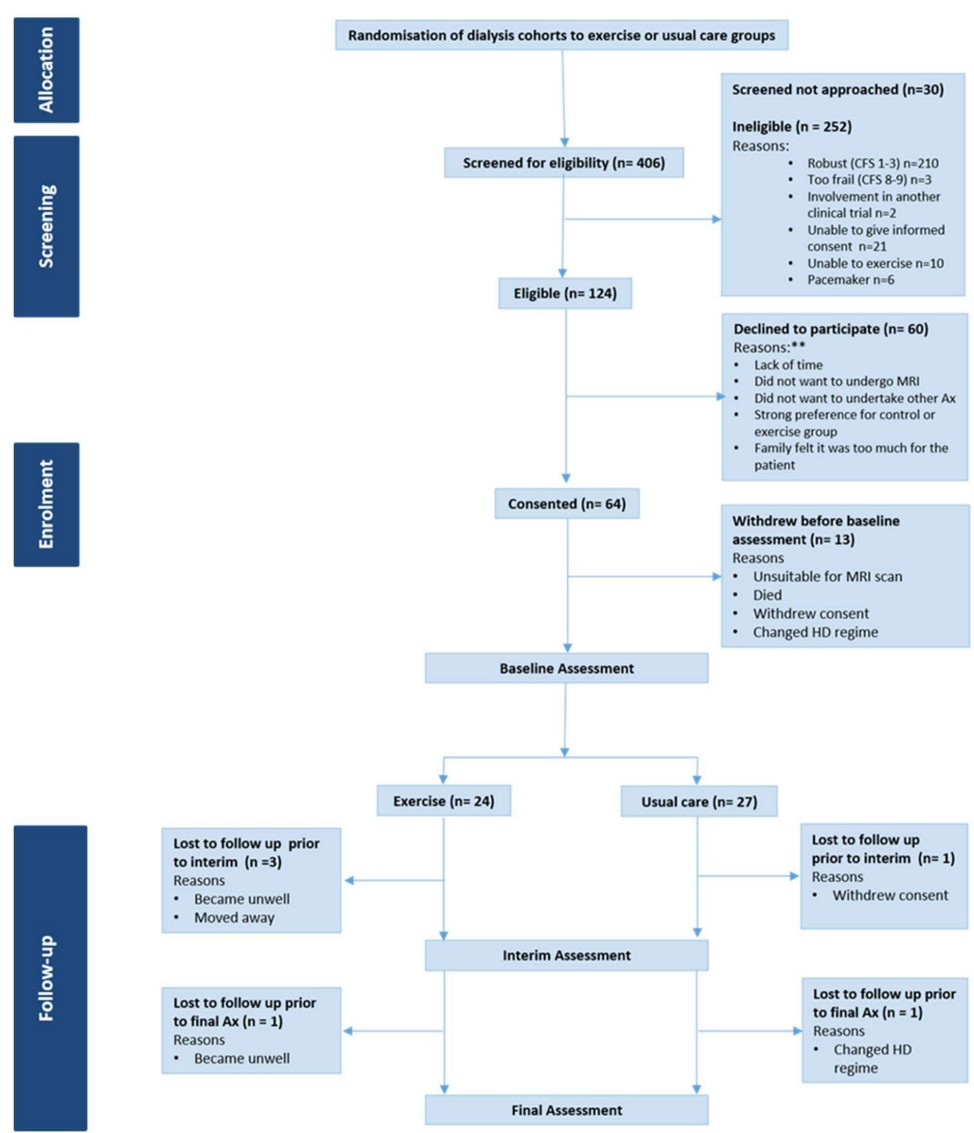

**Figure 1** CONSORT. Ax, assessment; CFS, Clinical Frailty Scale; HD, haemodialysis.** numbers not available, information only provided if freely given by participants.

## Retention

Six (12%, 95% CI 4.4% to 23.9%) participants were lost to follow-up: three participants withdrew due to ill-health, one moved away, one changed HD regime and one withdrew consent.

## Exercise adherence

A mean of 61±17 exercise sessions were completed over the 6-month intervention, representing an adherence rate of 74%±20%. The most frequent reasons for missing an exercise session were declining (n=175 out of 535 sessions omitted in total, 33%), feeling unwell (n=116, 22%) and pain (n=105, 20%). Table 2 summarises the mean amount of exercise achieved. On average, participants reached the prescribed level of exercise by 6 months, although n=18 (75%) were unable to achieve this by the end of the 1 month run-in period.

## Outcome acceptability

For tests of exercise capacity (ISWT and ESWT); n=14 (27%) did not complete at least one test at baseline, n=30 (64%) at interim and n=26 (58%) at final. For tests of physical function; n=20 (39%) did not complete at least one test at baseline, n=33 (70%) at interim and n=30 (67%) at final. For PROMs; n=27 (53%) did not complete at least one questionnaire at baseline, n=27 (57%) at interim and n=40 (89%) at final. For PA data; n=21 (41%) were missing at baseline, and n=26 (58%) were missing at the final assessment. Declining was the primary reason for non-completion for all outcomes across all time points.

## Secondary outcomes

Summary falls data are presented in online supplemental material 6. The crude falls incident rate ratio (IRR) was 1.95 (95% CI 0.63 to 7.18), suggestive of an almost

**Table 1** Baseline demographic and clinical characteristics of the trial participants

| | | Usual care (n=27) | Exercise (n=24) | All (n=51) |
|---|---|---|---|---|
| Age (years) | | 65±11 | 59±13 | 63±12 |
| Sex (n, %) | Female | 5 (19%) | 7 (29%) | 12 (23.5%) |
| Ethnicity (n, %) | White | 12 (44%) | 11 (46%) | 23 (45%) |
| | Asian or Asian British | 11 (41%) | 11 (46%) | 22 (43%) |
| | Caribbean | 1 (4%) | 0 (0%) | 1 (2%) |
| | Other ethnic | 1 (4%) | 1 (4%) | 2 (4%) |
| | Not stated | 2 (7%) | 1 (4%) | 3 (6%) |
| Diagnosis (n, %) | Aetiology uncertain | 8 (29%) | 7 (29%) | 15 (29%) |
| | Diabetic nephropathy | 5 (19%) | 7 (29%) | 12 (23%) |
| | Glomerulonephritis | 5 (19%) | 3 (14%) | 8 (16%) |
| | Renal vascular disease | 3 (11%) | 2 (8%) | 5 (10%) |
| | Other diagnoses | 4 (15%) | 1 (4%) | 5 (10%) |
| | Chronic pyelonephritis | 2 (7%) | 1 (4%) | 3 (6%) |
| | Polycystic kidney disease | 0 (0%) | 2 (8%) | 2 (4%) |
| | Not recorded | 0 (0%) | 1 (4%) | 1 (2%) |
| CCI | | 5±2 | 5±2 | 5±2 |
| Previous transplant (n, %) | No | 21 (78%) | 18 (75%) | 39 (76.5%) |
| | Yes | 6 (22%) | 6 (25%) | 12 (23.5%) |
| Time on HD (months) | | 17 (7–53) | 13 (10–61) | 16 (8–53) |
| BMI (kg/m$^2$) | | 27.38±6.72 | 25.87±5.28 | 26.67±6.07 |
| Total no. medications | | 12±4 | 12±4 | 12±4 |
| Clinical information | Albumin (g/L) | 35.4±4.4 | 37.4±4.3 | 36.4±4.4 |
| | Haemoglobin (g/L) | 107±12 | 112±17 | 107±15 |
| Haemodialysis | URR (%)* | 74 (70–80) | 75 (58–79) | 74 (71–79) |
| | SBP (mm Hg) | 143±21 | 144±21 | 144±21 |
| | DBP (mm Hg)* | 65 (62–78) | 78 (69–86) | 76 (62–81) |
| CFS (n, %) | 4, vulnerable | 13 (48%) | 10 (42%) | 23 (45%) |
| | 5, mildly frail | 5 (18.5%) | 7 (29%) | 12 (23.5%) |
| | 6, moderately frail | 8 (30%) | 5 (21%) | 13 (25.5%) |
| | 7, severely frail | 1 (3.5%) | 2 (8%) | 3 (6%) |

Values reported are mean and SD (±), except for *median and IQR.
BMI, body mass index; CCI, Charlson Comorbidity Index; CFS, Clinical Frailty Scale; DBP, diastolic blood pressure; SBP, systolic blood pressure; URR, urea reduction ratio.

two-fold increased incidence of falls within the usual care group.

Exercise capacity was maintained in the exercise group, but deteriorated in the usual care group, resulting in an overall difference of 36 m (95% CI −12 to 84) in ISWT results and 181 s (95% CI −92 to 453) in EWST time. The time taken to complete the Sit to stand five repetitions test (STS5, a component of the SPPB) also increased in the usual care group (suggesting a deterioration in function), but was maintained in the exercise group, resulting in an overall difference of 5 s (95% CI −4 to 15) (see online supplemental material 7).

Step count increased in the exercise group resulting in an overall difference of 859 steps/day (95% CI −825 to

**Table 2** Mean (SD) exercise achieved per session over the 6-month duration of the intervention

| | |
|---|---|
| Duration (mins) | 35±8 |
| Speed (RPM) | 63±10 |
| Intensity (RPE) | 13±1 |
| Gear | 9±4 |
| Distance (Miles) | 7±3 |
| Power (Watts) | 13±6 |
| Energy expenditure (Kcals) | 64±31 |

All data presented as mean and SD (±).
Kcals, kilocalories; mins, minutes; RPE, rating of perceived exertion; RPM, revolutions per minute.

2543) on HD days and 888 steps/day (95% CI −84 to 1861) on non-HD days. While sedentary time was increased in the exercise group on all days compared with the usual care group, this appeared to be offset by increases in light PA and moderate PA, and maintenance (although of low levels) of vigorous PA versus maintenance or deterioration across the same metrics in the usual care group (see online supplemental material 8). For PROMs, outcomes were largely unchanged, except for the Duke Activity Status Index (DASI) score, which appeared to deteriorate in the exercise group and increase in the usual care group, resulting in an overall difference in score of 4.93 (95% CI −0.94 to 10.80) and the mental component summary score of the SF12 which improved in the usual care group, resulting in an overall difference in score of 4 (95% CI −3 to 10). Exercisers appeared to have a greater perception of the benefits of exercise compared with those in the control group (3, 95% CI −4 to 11) (see online supplemental material 9).

### Serious adverse events

In total, n=13 (25%) experienced an SAE during the feasibility study, n=8 (33%) in the exercise group and n=5 (19%) in the usual care group. The most common reasons for SAEs were vascular access complications (n=3, 17%), stroke (n=3, 17%), acute coronary syndrome (n=2, 11%) and non-specific chest pain (n=2, 11%). All events were classed as serious as they resulted in hospitalisation. All resolved, and none were directly related to the intervention or trial.

### Qualitative findings

Thirty-seven patients were approached for the qualitative study. Twenty-six were recruited and one died prior to data collection. Thirteen had participated in the feasibility trial. Nine received dialysis during shifts randomised to exercise, and four during shifts randomised to usual care. Twelve participants had declined to take part in the feasibility trial. Full characteristics for the qualitative sample are provided in online supplemental material 10.

In addition to categories relating to the feasibility outcomes, categories relating to both the delivery and the characteristics of a tailored exercise intervention were identified.

### Feasibility and acceptability of a definitive trial
#### *Eligibility and recruitment*

Declining to participate was underpinned by a perception that the trial could worsen overall health, particularly among those who had not previously participated in research or had recently commenced HD. Female participants believed that exercise was predominantly for men and that they were already doing enough daily activity, while participants living with moderate to severe frailty viewed ageing as an inevitable decline unlikely to be influenced by exercise. Motivators included a sense of altruism, and the perception that participation could provide opportunities to improve individual outcomes;

learn about their own health and access better healthcare. Participants felt that recruitment could be enhanced by the effective use of non-verbal communication, rapport building, adaptation to study documentation and actively involving family members in the recruitment process, as family support was often a prerequisite to participation (table 3).

#### *Trial retention*

The primary reasons for withdrawal were becoming unwell, the duration of the trial and the research not meeting participants expectations. Participants suggested that having a rapport and maintaining regular dialogue with the research team might help retain participants within a future trial (table 3).

#### *The acceptability of IDC*

IDC was generally perceived to be a safe and positive use of HD treatment time. However, it was also described as limited in scope, and participants were uncertain of its impact, particularly on mobility, symptoms and falls (table 4).

#### *Outcome acceptability*

As indicated by participant quotations in table 5, the number of outcomes and follow-ups needed to be reduced and participants had a strong preference for outcomes that could be collected during HD treatment. Many found the ISWT and STS60 assessments too challenging. Participants were occasionally uncertain of the purpose of the questionnaires and many reported difficulty quantifying symptom severity or a desire to provide 'anticipated' responses.

Maintaining mobility, and the ability to undertake a range of ADLs and social roles were viewed as key outcomes for a future trial. Only 13 (52%) participants in the qualitative study agreed to complete a falls diary and many reported they preferred falls information to be collected during HD treatment. The majority who had fallen rarely reported them to healthcare professionals, believing that they were an expected consequence of HD or having experienced their concerns about falls being overlooked. Consequently, falls prevention was not viewed as a key outcome.

### Perceptions of a tailored exercise programme
#### *Delivery*

There was no universally acceptable setting for exercise delivery (table 6). Vulnerable and mildly frail participants (CFS 4–5) were particularly open to group-based exercise in the community or gym, which they felt would provide motivation through camaraderie with others. However, access barriers due to HD treatment, complex health needs and lack of transport were common. Participants also described feeling self-conscious exercising among 'normal' people. Home-based exercise was preferred by those with moderate to severe frailty (CFS 6–7) due to easier access, greater flexibility and relevance to their daily activities. Despite this, concerns about lack of space

**Table 3** Categories relating to trial eligibility, recruitment and retention with illustrative quotes

**Eligibility and recruitment**

| | |
|---|---|
| Challenges to recruitment | ► (Interviewer): 'Have you ever taken part in any research before?' (Participant): 'No. I have not been asked really'. (Female, moderately frail) |
| | ► 'If anything happens I am in trouble, I would rather avoid it (research)'. (Male, severely frail) |
| | ► 'I don't think I had been dialysing all that long and I didn't know how (the trial) would affect me'. (Male, mildly frail) |
| | ► 'I do enough, I am always out, up-down, do this, do that, I have just put clothes in the machine you know for a wash, I go for a shower you know'. (Female, mildly frail) |
| | ► 'You have got to take age into consideration. Now I am getting old and there is a limit to what I can do. And it doesn't get any easier it gets worse'. (Female, moderately frail) |
| Motivators to participation | ► 'If it helps someone else who has the same problem as me, they might be able to do something for him that they couldn't do for me'. (Male, moderately frail) |
| | ► 'I found the (outcome measures) very beneficial actually…it kind of educated me at the time…educationally it was informative'. (Male, vulnerable) |
| | ► 'What I like about research is that you are better looked after. I think if patients were a bit more aware that you are going to get preferential treatment, I think it would make it more attractive'. (Female, vulnerable) |
| Suggested methods of enhancing recruitment | ► 'The research team should be there and explain that they don't want much, explain the benefits. Explain it's not for us (the research team) it's for the patients benefit, let them try and if then it doesn't go well (the participant) can stop it… it's not the information you give but talking as a person that's more important'. (Male, vulnerable) |
| | ► 'If I have got confidence in (the researcher) and that (they) know what they are doing and why, then it's fine'. (Male, mildly frail) |
| | ► 'I don't like it (the text) is too tiny, I can't even read (the information sheet) with reading glasses on…a picture or two might also help'. (Female, mildly frail) |
| | ► 'There's a lot of sheets in (the information sheet), I think people will get fed up reading all that'. (Female, mildly frail) |

**Trial retention**

► 'I have thought of dropping out because I am unable to do much. I am not interested because…I am not well. I have got a lot of things (wrong) with my body'. (Female, mildly frail)

► 'Somebody recently asked me about research and I tried it for about three weeks and I said no, not for me…I thought no, this is not what I want, it's not particularly helpful'. (Female, moderately frail)

and safety were highlighted by those who lived alone, while those with family were concerned about overburdening or injuring them by asking for support.

**Characteristics of a tailored exercise programme**

Irrespective of the setting for delivery, participants

**Table 4** Categories relating to the acceptability of IDC and illustrative quotes

| | |
|---|---|
| A safe and positive use of HD treatment time | ► 'Yes, I found it useful. It made me do some exercise instead of just laying here drinking tea and watching TV, doing jigsaw puzzles'. (Male, mildly frail) |
| | ► 'They bring the bike but first they test you…whether you're safe to do it and all that.' (Female, mildly frail) |
| Limited scope and uncertain impact of IDC | ► 'We did cycling, and that was no choice because that's the only exercise we can do with our legs. You can't do sit-ups or stand-ups while you are lying down because you've got this thing (HD) going on' (Male, moderately frail) |
| | ► 'I thought maybe it helps, I get rid of some problems or maybe you know I am not walking too much…so I say maybe if I do start cycling…you know I can walk…but nothing happened, no nothing'. (Male, severely frail). |
| | ► 'My legs have become stronger, they were wobbly…it's more sturdy now than before. Yet I still have the falls, that I cannot help. But my legs are stronger than they were. I am a bit more agile than I used to be'. (Male, moderately frail) |
| | ► 'It was fine, it was ok, I got on with it. I used to have a laugh but then eventually my knees were just so painful then my (blood) pressure played up a bit'. (Female, vulnerable) |
| | ► 'Blood pressure was coming down. I used to take medication for the blood pressure now I don't take it'. (Male, vulnerable) |

HD, haemodialysis; IDC, intradialytic cycling.

**Table 5** Categories relating to outcome acceptability and illustrative quotes

| | |
|---|---|
| Perceptions of outcome assessments | ► 'It was a bit of a task, too many (outcomes) personally'. (Male, mildly frail)<br>► 'It's really helpful if it's (outcome assessment) done here whilst I am on dialysis. We have got all this free time. Sometimes its five medical appointments a week, Tuesdays and Thursdays (non-dialysis days) become quite precious to me'. (Male, vulnerable)<br>► 'The walking ones (tests) I could make the distance, but the time was ridiculous, they asked me to do it fast. I can't, I have only got one speed'. (Male, vulnerable)<br>► 'I am not very good at scores, or you know, what they say about pain, what number it is? I am no good at that. I don't know what it means. I know it really hurts but I just can't describe the extent of it. It's difficult to put it in a number like that'. (Female, vulnerable)<br>► 'Like all form filling, you can be undecided as to what or how to answer them. Sometimes you don't, you kind of guess what you should be saying'. (Male, mildly frail) |
| Important outcomes | **Maintaining mobility**<br>► 'If you are walking better you are not getting out of breath and that's what does me. I mean I can't walk down this corridor to the ambulance because I am having to stop and get my breath back'. (Female, moderately frail)<br><br>**Maintaining activities of daily living and social roles**<br>► 'I don't want to walk miles I just want to do enough to get around…from my chair to my commode or from my commode onto the bed. The only way I can do that is with the rotunda at the minute. I would like to do it with my walking frame'. (Female, moderately frail)<br>► 'I just want to carry on living and enjoying my life with (my partner) and children, my sisters, and of course all my friends, the church involvement, because I want to enjoy that for absolutely as long as I can'. (Male, mildly frail)<br>**Falls and falls diaries**<br>► 'I don't fall on a weekly basis… falling over is not something that happens on any sort of regular basis'. (Male, moderately frail)<br>► 'When I was at the hospital, I told them I had a fall. They don't want to know. They said, "You are perfect, your levels (bloods) are perfect and everything"'. (Male, vulnerable)<br>► 'You know I sometimes I forget (to write in the diary). So, the first days I had written and then I forgot it. And when you forget it then you can't get the information right'. (Male, severely frail)<br>► 'I can't hold a pen properly, so I am not able to write. (Because of) arthritis they said, because I have got neuropathy and because I am on dialysis phosphate is causing my fingers to sometimes…close up'. (Male, moderately frail)<br>► 'If (the researcher is) opposite you and gives you the information,(they're) going to explain it even better, you know (they) can even ask (the participant) what happened and then they explain to (the researcher) different. But you forget you know the diary it's very difficult and some of (the participants) won't ever to know how to use it'. (Male, mildly frail) |

identified several key features of a tailored exercise intervention which are summarised in figure 2.

*Preparation*

Participants lived with a range of debilitating symptoms, most frequently fatigue, pain and dyspnoea. Often daily activity alone was felt to be enough of a challenge. Common impacts of exercise (eg, breathlessness while exercising) were interpreted as worsening symptoms or damage, and many participants were uncertain if exercise would be suitable or beneficial. They indicated that the reason for exercising needed to be sufficiently compelling. They wanted to know what to expect prior to exercising, and individualised goal setting was advocated to build motivation and appreciate improvements.

*Content*

Key components described were whole body resistance, aerobic and balance training. Many participants described being unable to get up once they had fallen and felt that practising this was also important. Routine PA was viewed

as more purposeful than structured exercise 'for the sake of it' and participants spoke of their enjoyment of being outside and engaging in meaningful and physically active hobbies.

*Structure*

Supervision was viewed as essential to select, teach and progress exercises. Individual tailoring which considered the impact of disability, comorbidities and fluctuating symptoms was important, and a choice of exercises, for example swimming, dancing and yoga, was associated with increased enjoyment and engagement. Moderate to severely frail participants wanted the programme to be progressed in a supportive and collaborative manner. Those who were vulnerable or mildly frail wanted to be 'pushed' and progressed in a more assertive manner.

Having a companion (typically peers, family or friends) was viewed as helping to overcome access barriers and provide socialisation and mutual motivation. The sharing of experience was also seen as a powerful means of

**Table 6** Participants perceptions of the facilitators and barriers to group and home-based exercise

| Exercise setting | Facilitators | Barriers |
|---|---|---|
| Group community or gym-based exercise | 'There is something about the group dynamics. When you try and do it on your own you can't really focus. It's just so much easier to do as a group than an individual, especially if you have got motivational problems and you're having to do this (dialysis)'. (Male, vulnerable) | 'I was lucky enough that my wife was off so she took me and brought me, otherwise transport was a problem, sometimes I used to take a taxi because hospital transport you can't trust it'. (Male, moderately frail) |
| | 'Better to be in a group, because when you see other people doing it, you just automatically join in and you feel like she can do it why not me?' (Female, mildly frail) | 'I have only got Tuesday and Thursday and most of the days that cropped up (to attend a falls prevention programme) they are either on a Wednesday or a Friday when I couldn't go because I have dialysis'. (Male, mildly frail) |
| | 'We are all in the same boat. You can say "How are you going on this week? You know you are on dialysis, are you finding this OK?" and you can get notes from them'. (Female, severely frail) | 'Apparently because of my complex problems and disabilities he (participants GP) doesn't think anyone at the gym is sufficiently qualified to tell me which exercises are best'. (Female, moderately frail) |
| | | 'I would love to go to the gym and start sorting myself, but it's just a normal gym where normal keep-fit people go, so I have never ended up there'. (Female, vulnerable) |
| Home-based | 'When you are at home exercise is normal it really is. If you are going upstairs to get something you don't think…I am not going up there to get that. You go upstairs and get it because that's part of your everyday life'. (Male, moderately frail) | 'It's just the room that you have got where you can do exercise…if you haven't got that it's very difficult'. (Female, moderately frail) |
| | | 'I can't do anything in the home. There is no-one there, I'm alone, what if anything happens?' (Male, mildly frail) |
| | | 'I am nervous about practising at home because if I couldn't get up, I don't want my husband hurting his back. I shall have to wait until a friend comes around and they could both help me'. (Female, moderately frail) |

challenging preconceptions about exercise ability, although participants with moderate to severe frailty raised concerns about feeling embarrassed or 'judged' if they were less able.

### Integrated mixed-methods analyses

The integrated qualitative and quantitative findings suggest that an RCT of IDC is feasible for frail HD patients following adaptation. However, IDC should not be the only intervention offered and the development of a multicomponent programme is warranted (see online supplemental material 11).

### DISCUSSION

These results suggest that an RCT of IDC is feasible for frail HD patients with adaptation to increase outcome acceptability and eligibility rates. Adherence to IDC was high and it was viewed as a safe and efficient use of HD treatment time. Secondary outcomes also suggest that, for HD patients with a CFS of 4–7, IDC may mitigate deterioration in exercise capacity, endurance and functional muscle strength, increase PA behaviour (steps/day) and reduce falls incidence. Despite this, participants described a preference for a multicomponent programme that

prepared them for exercise, offered variety, companionship and individualised supervision. No single preferred environment for the delivery of this intervention was identified, but appeared to be influenced by frailty grade and individual factors.

27%–89% of secondary outcome measure data were missing, and, overall, this progression criterion was not achieved. Given that secondary measures are often insufficiently powered, reducing the number collected within a future trial may improve completion.[26] Falls were not of primary importance to participants, and this aligns with SONG-HD data which did not identify falls as a key outcome.[27] Our findings suggest that accurately capturing prospective falls data may be challenging due to under-reporting, and yet, retrospective falls data collection does not fully reflect the incidence and impact of falls, particularly those which do not require an ED visit or hospital admission. Given the high incidence of falls in this population, capturing falls data may be important in a future trial, and regular prospective recording of information relating to falls as a part of routine practice at the dialysis unit is recommended, in line with participant feedback.[5] This would provide both clinicians and researchers with

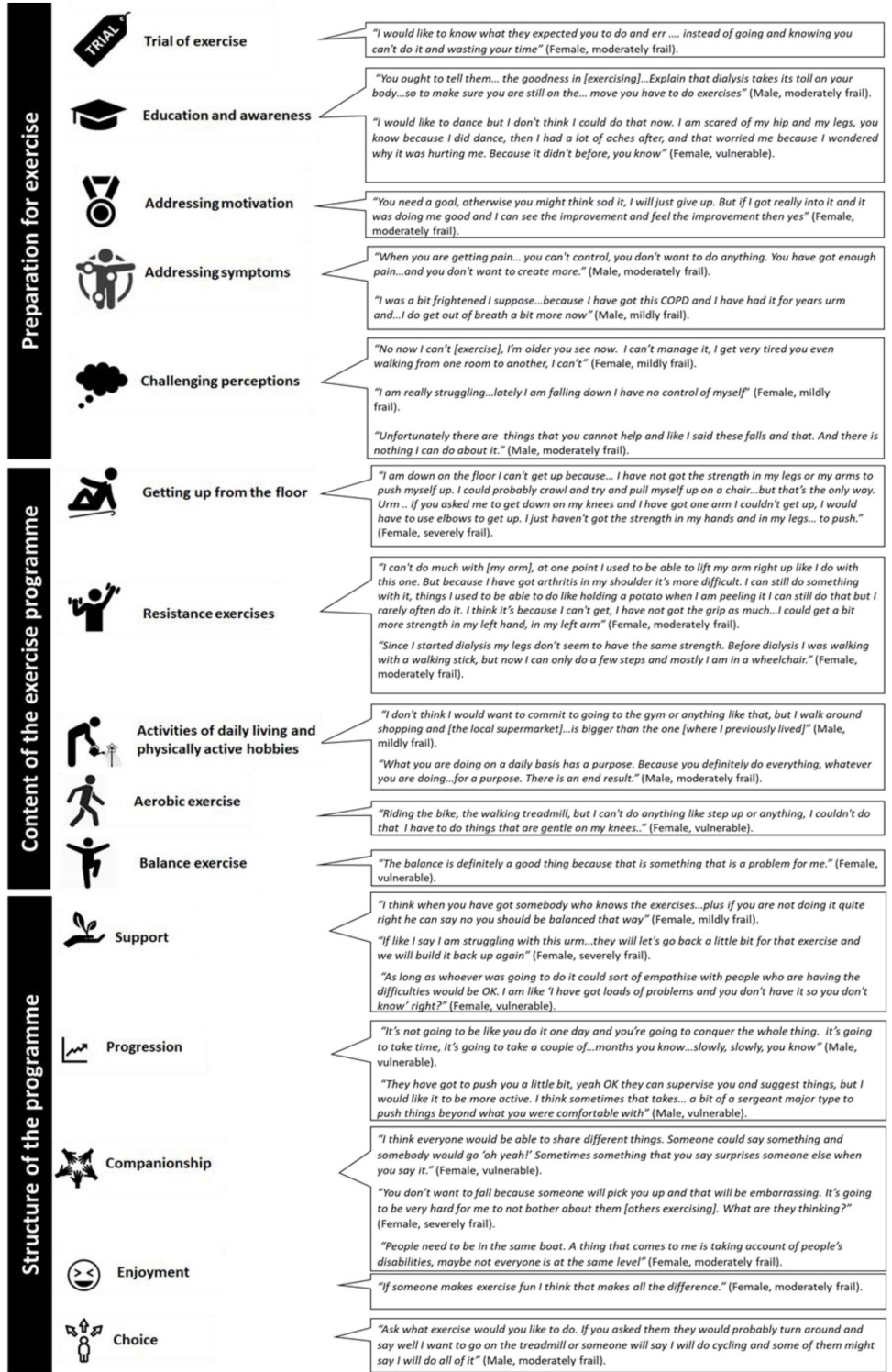

**Figure 2** The core components of an acceptable exercise programme for people living with frailty and receiving haemodialysis.

higher quality data for use in both prospective and retrospective studies, and to inform clinical care.

Further exploration and validation of meaningful measures for HD patients living with frailty is also warranted. Some of the functional measures (the STS60 and ISWT) included were too challenging. In the absence of a core set of functional outcome measures for older people, or people receiving HD, we suggest

that the SPPB may be the most appropriate and feasible method of capturing information about mobility and function. Although challenges with ceiling effects have been identified, this measure had the lowest levels of non-completion within this study, and has demonstrated good test-retest reliability in HD patients and excellent validity and responsiveness to change following an intervention in older adults.[28 29] To date, measures of basic and

instrumental ADL ability and participation have rarely been used in exercise studies in an HD population. These outcomes were, however, highlighted as important within this study, and have also been included in guidelines and core outcomes sets for HD and older people, warranting their inclusion in future exercise studies relating to frail HD populations.[27 30 31]

The results of this study indicate that changes to eligibility criteria and screening processes are required. As only patient participants were interviewed, it was not possible to gain any insight on this aspect of feasibility from the qualitative component. Importantly, the challenges of identifying eligible participants do not appear to be unique to this study and a multicentre trial may be required.[32]

Higher proportions of older, female and more severely frail HD patients declined to participate and the qualitative data indicated this was due to negative perceptions relating to participation in both exercise and research. Such findings clearly have implications for the external validity of a future trial and the reach of the intervention at the point of implementation.[24] To address this, this study suggests recruitment strategies which use effective non-verbal communication skills to build rapport and explore participants' perceptions of the intervention and the research process, and subsequently provide balanced information about the study, may lead to more representative recruitment. A sense of equipoise may be preserved by emphasising altruism, access to potentially enhanced care, and an opportunity to learn about their health (which were all identified as motivators to participation), rather than the potential individual benefits of the intervention itself. Involving families and/or peer supporters who have experience of the study and intervention in the recruitment process and introducing opportunities for participants to observe the exercise intervention may also be beneficial. Ultimately, the selection of these strategies will depend on the resources available and the need to strike a balance between conducting a trial with high internal and external validity and going beyond what is pragmatically possible to engage patients in the intervention at the implementation phase.

This study suggests that IDC may reduce the incidence of falls resulting in ED visits and hospital admissions in frail HD patients potentially by attenuating a decline in exercise capacity, PA behaviour and function at levels shown to be clinically meaningful in other long-term conditions.[33 34] This indicates that preventing deterioration may be as valuable, and more attainable, as improving outcomes in a frail population. Despite this, frail participants experienced difficulties achieving the proposed level of exercise and maintaining motivation in the face of varying symptomology. Exercise programmes have a dose–response, and these factors may have reduced participants physical capability to exercise and achieve optimal benefit, despite the overall good level of adherence. Clinical decision support tools have been used in other populations to rationalise exercise prescription,

progression and amendment in the presence of varying symptomology and a similar approach may be beneficial for frail HD patients.[35]

This study indicates that participants desire a multicomponent exercise programme, and require an intervention that addresses their particularly low levels of PA. While step count and time spent in light and moderate PA increased following IDC, these were below PA recommendations for older people.[36] To date, PA interventions for HD patients have predominantly centred around walking, which may not be appropriate for those living with frailty.[37–40] This study suggests that functional training (task-orientated exercise which engages multiple muscle groups) and PA that focuses on 'doing more' of these usual tasks may be more acceptable and efficacious. To date, two studies have employed similar approaches with non-frail HD patients. One study demonstrated significant improvements in lower extremity performance and the other a non-significant improvement in physical function and maintenance of other domains within the 36-Item Short Form Health Survey (SF-36) compared with the control group.[41 42] In older people without CKD who are living with frailty, functional training included as part of a multicomponent exercise programme is beneficial across a range of outcomes, including greater ability to rise from the floor following a fall.[40 43–46] A similar approach to exercise prescription may be warranted in a frail HD population.

Numerous barriers and facilitators to exercise were identified within this study, which have implications for the design of a programme. The use of theory is crucial in the development of effective interventions and the behaviour change wheel (BCW) is most frequently cited in the development of interventions in CKD.[47] Mapping the identified barriers and facilitators to the BCW indicates that ameliorating symptom burden prior to exercise, individualised exercise counselling and a collaborative, problem-solving approach to exercise education are most likely to encourage and sustain participation.[47 48] Devising ways in which peer and family involvement can be incorporated into the programme may also increase motivation and opportunity to exercise but should be carefully managed given the potential for negative comparison among the frailest patients.

A lack of preferred environment for intervention delivery may have implications for a definitive RCT. Exercise interventions require motivation, and limited engagement may negatively influence a trials external and internal validity. Ignoring patient preference is also out of step with clinical practice, where rehabilitation involves shared decision-making. Taken together, these factors have implications for determining treatment effects and future intervention implementation.[49] There is increasing recognition that novel trial designs may be indicated when evaluating complex interventions and a Partially Randomised Patient Preference Trial, where participants without preference are randomised, while those with a preference receive their choice would

provide information on both the efficacy of the intervention and the influence of preference.[49 50]

## Strengths and limitations

To the best of our knowledge, this study is the first to examine the feasibility of an RCT of IDC for frail HD patients and to explore how trial procedures and exercise programmes should be specifically tailored to the needs of this group, from their own perspectives. Key strengths were the use of a validated frailty risk-stratification measure and multiple qualitative methods which provided a form of triangulation.[51] There were, however, challenges to recruiting severely frail participants, and those from a more diverse range of black and minority ethnic groups, to both the trial and the qualitative study. Additionally, the views of clinicians and researchers were not explored. A future RCT should also blind outcome assessors to group allocation to reduce the potential for detection bias. Finally, this study is exploratory and therefore all secondary measures of exercise capacity, function and PROMs should be interpreted with caution, not least due to the high number of participants who did not complete the follow-up tests.

## CONCLUSION

In summary, this study suggests that a future definitive trial of IDC is feasible within a HD population with a CFS of 4–7 and paying particular attention in the design to those factors mentioned above may facilitate improved rates of eligibility and outcome completion. Outcomes focusing on independence and participation should be the primary outcomes of interest in a future trial. While an exploratory analysis suggests some potential benefits to IDC, a tailored intervention comprising a comprehensive multicomponent exercise programme, symptom management, education and behaviour change is better suited to frail HD patients' needs.

## Author affiliations

[1]Department of Respiratory Sciences, University of Leicester, Leicester, UK
[2]Department of Research and Innovation, University Hospitals of Leicester NHS Trust, Leicester, UK
[3]Department of Cardiovascular Sciences, University of Leicester, Leicester, UK
[4]National Centre for Sport and Exercise Medicine, Loughborough University, Loughborough, UK
[5]Department of Health Sciences, University of Leicester, Leicester, UK
[6]Emergency Department, Leicester Royal Infirmary, University Hospitals of Leicester NHS Trust, Leicester, UK
[7]Renal, Respiratory and Cardiovascular Clinical Management Group, University Hospitals of Leicester NHS Trust, Leicester, UK
[8]Department of Physiotherapy and Renal Medicine, King's College Hospital, London, UK
[9]Department of Renal Medicine, King's College London, London, UK
[10]Usher Institute, University of Edinburgh, Edinburgh, United Kingdom
[11]Centre for Exercise and Rehabilitation Science, Leicester Biomedical Research Unit, Leicester, UK
[12]Department of Respiratory Medicine, Glenfield Hospital, University Hospitals of Leicester NHS Trust, Leicester, UK

**Acknowledgements** The authors would like to thank the staff and patients at all units, Fresenius Medical Care and The Leicester Kidney Care Appeal. The authors also wish to thank Freya Tyrer for sharing her statistical expertise.

**Contributors** Literature search: HMLY; research idea and study design: HMLY, JOB; participant recruitment: HMLY, DSM, PJH, DCC, MPMG-B, JOB, CG; data acquisition: HMLY, DSM, PJH, DCC, MPMG-B, CG, WJ, M-MC; clinical governance: JOB; data analysis: HMLY, SG; statistical analysis: HMLY; supervision and mentorship: SC, HCE, SAG, SJS, ACS, JOB; manuscript preparation: HMLY; reviewed final manuscript: all. Each author contributed important intellectual content during manuscript drafting or revision and accepts accountability for the overall work by ensuring that questions pertaining to the accuracy or integrity of any portion of the work are appropriately investigated and resolved. All authors have read and approved the final version.

**Funding** The work was funded by the National Institute for Health Research (NIHR) Leicester Biomedical Research Centre, Collaboration for Leadership in Applied Health Research and Care East Midlands (CLAHRC EM) and partly funded by the Stoneygate Trust. HY and JOB are supported by grants from the NIHR (DRF-2016-09-015 and CS-2013-13-014). SJS is supported by the Collaboration for Leadership in Applied Health Research and Care East Midlands. The views expressed in this publication are those of the authors and not necessarily those of the NHS, the NIHR or the Department of Health and Social Care. The funders had no role in the study design; collection, analysis and interpretation of the data.

**Competing interests** None declared.

**Patient consent for publication** Not required.

**Ethics approval** This study was approved by the East Midlands (Northampton; REC ref: 14/EM/1190) and South West (Bristol; REC ref: 17/SW/0048) NHS Research Ethics Committees for the trial and the qualitative component respectively.

**Provenance and peer review** Not commissioned; externally peer reviewed.

**Data availability statement** The datasets used and analysed during the current study are available from the corresponding author on reasonable request.

**ORCID iD**
Hannah M L Young http://orcid.org/0000-0002-4249-9060

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
