## [Reviewer comments · BMJ Open]

ARTICLE DETAILS

TITLE (PROVISIONAL)	Exercise for people living with frailty and receiving haemodialysis: a mixed-methods randomised controlled feasibility study.
AUTHORS	Young, Hannah; March, Daniel; Highton, Patrick; Graham-Brown, Matthew; Churchward, Darren; Grantham, Charlotte; Goodliffe, Samantha; Jones, William; Cheung, Mei-Mei; Greenwood, Sharlene; Eborall, Helen; Conroy, Simon; Singh, Sally; Smith, Alice; Burton, James

VERSION 1 – REVIEW

REVIEWER	Adnan Sharif University Hospitals Birmingham United Kingdom
REVIEW RETURNED	25-Jun-2020

GENERAL COMMENTS	Thank you for asking me to review this manuscript from Young and colleagues, exploring results from a mixed-methods randomised controlled feasibility study involving exercise for people on haemodialysis living with frailty. The group has an established track record and the manuscript is clear in its structure and objectives. Frailty is also a topical issue for dialysis patients and the results of this feasibility work is informative for the community. My minor comments are; 1. It may be worth adding to Table 1 the corresponding CFS score for the different sub-groups for clarity2. Was there a difference in demographics for people who were willing to participate versus those who were not or dropped out? For example with regards to age, sex and/or ethnicity?3. Was a follow up CFS score conducted at the end of the intervention? Would be interesting to see if there was any improvement? Although the caveat being, as a non-blinded intervention, there would be a significant element of subjective bias and the more objective findings are therefore more of value.4. The qualitative study components are really interesting. I would be keen to know if participants were asked their opinion as to what frailty means to them? I assume it has different connotations for different individuals. It may be important when planning for frailty-related outcomes.
---

REVIEWER	Danielle Kirkman Virginia Commonwealth University, USA
-----------------	---

REVIEW RETURNED	29-Jun-2020
-------------

GENERAL COMMENTS	Whilst there is a wealth of evidence documenting the health related effects of intradialytic exercise, this study is unique as it is the first study to investigate the feasibility of this therapeutic strategy in a frail cohort of hemodialysis patients. The results suggest that a randomized controlled trial (RCT) of intradialytic cycling exercise for frail HD patients is feasible. Based on their findings, the authors suggest edits to their current protocol towards implementing a more effective intervention and study design in future RCTs. A very important and informative study, the authors should be commended for completing this feasibility trial constituting both qualitative and quantitative analyses. Both the quantitative and qualitative aspects have sound scientific design. Minor comments:  - In the discussion it is difficult to interpret the results of IDC on exercise capacity as beneficial due to the extremely high number of patients who did not complete the follow up tests. It is suggested that this interpretation is discussed with caution. - In line 451 - 454 perhaps some suggestions for future trials of more feasible physical function tests for these patients could be included. - It is understood that the recruited sample is representative of the trial center communities. However, the low representation of black individuals and females (sex as a biological) could be added as an additional limitation of the dataset. - In Table 1 it is not clear why there is inconsistency in data reporting i.e. some variables are presented as mean and standard deviation and others as interquartile range. - It would be helpful if the figures were referenced in the main body (eg. CONSORT flow diagram).
--

REVIEWER	Tobia Zanotto University of Illinois at Urbana-Champaign, U.S.A.
REVIEW RETURNED	10-Jul-2020

GENERAL COMMENTS	This is a well written manuscript on the feasibility of an intradialytic cycling (IDC) exercise intervention for frail haemodialysis (HD) patients. The methodological approach is appropriate and findings have been thoughtfully discussed. I have a few (minor) comments which could further improve this paper:  - Serious adverse events: can you comment on what types of SAE were recorded during the study? - Table 1 (participant characteristics): please provide the p-values. This information would be helpful to better interpret the matching between groups. Also, please clarify how the frailty categories (e.g. "vulnerable", "mildly frail", "moderately frail", "severely frail") were defined based on the CFS scores. - Operational definition of frailty: 1) Can you elaborate on why you chose the CFS to screen frailty (i.e. why this operationalisation vs other frailty measures such as the Fried's frailty phenotype, frailty index etc.)? 2) Would it be accurate to assume CFS-4 is actually more comparable to a "pre-frailty" status? 3) In the Discussion, you
--

	should acknowledge that interpretation of findings is partially contingent on the current operationalisation of frailty. - Operational definition of falls: 1) The decision of prospectively recording falls with visit to emergency department is methodologically sound (i.e. objective and easily-traceable data). However, the large majority of falls does not result in severe injury (and thus in A&E visit) and you should mention that most non-injurious falls have not been captured in your study (during follow-up). 2) It is also unclear how many participants experienced falls during the prospective follow-ups (i.e. how many single- and recurrent-fallers?). 3) Please clarify how the crude falls incidence rate ratio was calculated. - Discussion, lines 448-450: I agree with your point that although falls diaries (completed by patients) are usually regarded as the gold standard for falls monitoring, in the context of CKD-5-HD, prospective data collection at the dialysis unit may be a more suitable approach. Perhaps you can take one step more and suggest that recording falls information routinely during the HD session (e.g. once a week) may be useful in clinical practice and would also allow researchers to rely on higher quality falls data (for both prospective and retrospective studies).
--	--

VERSION 1 – AUTHOR RESPONSE

Reviewer 1

It may be worth adding to Table 1 the corresponding CFS score for the different sub-groups for clarity

Thank you for pointing this out, we agree and have added the CFS score to table 1 for clarity.

Was there a difference in demographics for people who were willing to participate versus those who were not or dropped out? For example, with regards to age, sex and/or ethnicity?

Thank you for your comment. We did look at this, but omitted it from the final manuscript due to word count constraints. We have now added this information back in on page 13, lines 255-259 as follows:

“Those who declined to participate had a median age of 73 (IQR 67-81) years. N=35 (58%) were female and n=27 (42%) male. Twenty-five (42%) were classified as vulnerable according to the CFS, n=17 (28%) were mildly frail, n=9 (15%) moderately frail and n=9 (15%) severely frail.”

This ties in with qualitative findings, particularly participants reasons for being reluctant to participate in the trial, so we have also added the following into the discussion on page 29, lines 497-515 in recognition of this:

“Higher proportions older, female and more severely frail HD patients declined to participate whilst the qualitative data indicated this was due to negative perceptions relating to participation in both exercise and research. Such findings clearly have implications for the external validity of a future trial and the reach of the intervention at the point of implementation.”²⁴

To address this, this study suggests the recruitment strategies which utilise effective non-verbal communication skills to build rapport and explore participants’ perceptions of the intervention and the research process, and subsequently provide balanced information about the study, may lead to more representative recruitment. A sense of equipoise may be preserved by emphasising altruism, access to potentially enhanced care, and an opportunity to learn about their health (which were all identified as motivators to participation), rather than the potential individual benefits of the intervention itself. Involving families and/or peer supporters who have experience of the study and intervention in the recruitment process, and introducing opportunities for participants to observe the exercise intervention, may also be beneficial. Ultimately the selection of these strategies will depend upon the resources available and the need to strike a balance between conducting a trial with high internal and external validity and going beyond what is pragmatically possible to engage patients in the intervention at the implementation phase”

Was a follow up CFS score conducted at the end of the intervention? Would be interesting to see if there was any improvement? Although the caveat being, as a non-blinded intervention, there would be a significant element of subjective bias and the more objective findings are therefore more of value.

We agree this could potentially be something that would be interesting to look at, however we did not undertake to do this within this study for the reasons that the reviewer identifies, but also because the CFS was originally developed for use as a risk stratification tool rather than an outcome measure. As such, it is unknown if a change in frailty status would translate into a clinically meaningful improvement.

The qualitative study components are really interesting. I would be keen to know if participants were asked their opinion as to what frailty means to them? I assume it has different connotations for different individuals. It may be important when planning for frailty-related outcomes.

Thank you for your comments on the qualitative component. The interview and diary data described in this manuscript focus entirely upon the trial and the exercise intervention, but we did conduct a separate interview with frail HD participants which focused on their wider experiences of living with frailty and their perceptions of renal care, and healthcare more widely. At the conception of the study the meaning they ascribed to frailty was something that we considered asking the participants within these interviews. However, following extensive discussion with our PPI group, we found that many people did not identify themselves as frail, although they could identify many of the characteristics of frailty, and the impact of these upon their daily lives. We were concerned this may prevent eligible people from taking part in the study and therefore gaining a diverse range of views. Because of this, and in light of a report from AGE UK (Frailty: Language and Perceptions, 2015, https://www.ageuk.org.uk/documents/en-gb/for-professionals/policy/health-and-wellbeing/report_bgs_frailty_language_and_perceptions.pdf), which highlights that using the term

'frailty' can provoke a strongly negative reaction from older people, we removed this line of questioning. Instead, we used understanding how the characteristics of frailty impacted their experience as a route into understanding how frailty influenced their experiences and perceptions of care. We hope to publish this work as a separate article in due course.

Reviewer 2

In the discussion it is difficult to interpret the results of IDC on exercise capacity as beneficial due to the extremely high number of patients who did not complete the follow up tests. It is suggested that this interpretation is discussed with caution.

We completely agree and have added the following to reflect this point on page 32, lines 579-581:

“Finally, this study is exploratory and therefore all secondary measures of exercise capacity, function and PROMS should be interpreted with caution, not least due to the high number of participants who did not complete the follow up tests.”

In line 451 - 454 perhaps some suggestions for future trials of more feasible physical function tests for these patients could be included.

Thank you, this is a good point and we have added the following information on page 28, lines 476-486:

“In the absence of a core set of functional outcome measures for older people, or those receiving haemodialysis, we suggest that the SPPB may be most appropriate and feasible measure for capturing information about mobility and function. Although challenges with ceiling effects have been identified, this measure had the lowest levels of non-completion within this study, and has demonstrated good test-retest reliability in HD patients and excellent validity and responsiveness to change following an intervention in older adults.^{28,29} To date, more comprehensive measures of basic and instrumental ADL ability and participation have rarely been used in exercise studies. These outcomes were, however, highlighted as important within this study, and have also been included in guidelines and core outcomes sets for HD and older people, warranting their inclusion in future exercise studies relating to frail HD populations.^{27,30,31}”

It is understood that the recruited sample is representative of the trial center communities. However, the low representation of black individuals and females (sex as a biological) could be added as an additional limitation of the dataset.

Thank you, we hope we have made the issues with the recruitment of female participants more explicit in responding to reviewer ones feedback relating to participants who declined, and have dedicated some more of the discussion to this, which is outlined above in our response to viewer one. In relation to the representation of other black and minority ethnic groups, our sample is (as reviewer two highlights) representative of the local population, but we appreciate that this may influence the

wider generalisability of the results and so have added the following on page 32, lines 574-576 within the limitations section:

“There were, however, challenges to recruiting severely frail participants, and those from a more diverse range of black and minority ethnic groups, to both the trial and the qualitative study.”

In Table 1 it is not clear why there is inconsistency in data reporting i.e. some variables are presented as mean and standard deviation and others as interquartile range.

Thank you for noticing this, apologies for omitting this from the statistical analysis section, but this is the approach preferred within CONSORT guidance. We have now clarified as follows on page 11, lines 218-219:

“Sample characteristics are presented as mean \pm standard deviation, median (IQR) or n (%), as appropriate.”

It would be helpful if the figures were referenced in the main body (eg. CONSORT flow diagram).

Thank you, figures are referenced within the main text (figure 1 on page 13, line 251 and figure 2 on page 18 and line 342) but we have also added place markers for them within the main body.

Reviewer 3

Can you comment on what types of SAE were recorded during the study?

Thank you for your comment, we have provided some further information on the types of SAEs as follows on page 17, lines 327-330:

“The most common reasons for SAEs were vascular access complications (n=3, 17%), stroke (n=3, 17%), acute coronary syndrome (n=2, 11%) and non-specific chest pain (n=2, 11%).”

Table 1 (participant characteristics): please provide the p-values. This information would be helpful to better interpret the matching between groups. Also, please clarify how the frailty categories (e.g. “vulnerable”, “mildly frail”, “moderately frail”, “severely frail”) were defined based on the CFS scores.

We have added the CFS scores alongside the descriptors in table one, as per your comments and the suggestions of reviewer one. Whilst we appreciate that adding p-values may help to indicate any differences between intervention groups, in accordance with CONSORT guidance, we feel it is inappropriate to report p values for the following reasons. Firstly, baseline imbalances are more likely

in a feasibility study due to smaller numbers of participants. Secondly, the differences observed are likely due to chance as randomisation would have helped to mitigate selection bias. Finally, it is less relevant to a feasibility study. We believe that we have presented the baseline data in such a way that allows the reader to interpret the characteristics of the groups and the size of any chance imbalances that have occurred.

Can you elaborate on why you chose the CFS to screen frailty (i.e. why this operationalisation vs other frailty measures such as the Fried's frailty phenotype, frailty index etc.)?

We chose to use the CFS because the scale has good predictive abilities in and HD populations and good construct validity when compared with the Frailty Index (Rockwood et al. A global clinical measure of fitness and frailty in elderly people. *CMAJ*. 2005;173(5):489-495 and Alfaadhel et al. Frailty and mortality in dialysis: Evaluation of a clinical frailty scale. *Clin J Am Soc Nephrol*. 2015;10(5):832-840). It has also been validated in the HD population and is subsequently gaining traction within clinical practice and research (Nixon et al. Diagnostic accuracy of frailty screening methods in advanced chronic kidney disease. *Nephron*. 2019;141(3):147-155). Whilst the phenotype has become the *de facto* gold standard measurement of frailty in HD studies, we chose not to use this our PPI group felt this would have created an additional burden for potential participants at the screening stage. Some authors have opted to substitute self-reported measures for the original objective assessments in light of this, however we opted not to do so because of the risk of significantly overestimating the prevalence of frailty within our population. Additionally, whether it is appropriate to apply the thresholds identified within the Fried et al (2001) study (which focused on a community dwelling older population) to an HD population has not yet been established. However, considering your comment we have elaborated on our rationale for the use of the CFS on page 7, lines 132-134.

"The CFS has good predictive abilities in an HD population, good construct validity when compared with the Frailty Index, is less burdensome than the Frailty Phenotype, and has been validated in an HD population,"

Would it be accurate to assume CFS-4 is actually more comparable to a "pre-frailty" status?

This is a reasonable assumption, the operational definition for CFS 4 is "whilst not dependent on others for daily help, often symptoms limit activities. A common complaint is being 'slowed up', and/or being tired during the day" (Rockwood et al. A global clinical measure of fitness and frailty in elderly people. *CMAJ*. 2005;173(5):489-495). We included a vulnerable population within the study sample because these participants are at increased risk of developing frailty and represent a group of patients who have the potential to benefit from an exercise intervention.

In the Discussion, you should acknowledge that interpretation of findings is partially contingent on the current operationalisation of frailty.

For the reasons outlined in our response above justifying the choice of the CFS and outlining some of the potential limitations of the use of the phenotype in an HD population, we do not believe that it is a

limitation of our work. The CFS is a validated risk stratification tool in an HD population and has been used within numerous HD studies.

The decision of prospectively recording falls with visit to emergency department is methodologically sound (i.e. objective and easily-traceable data). However, the large majority of falls does not result in severe injury (and thus in A&E visit) and you should mention that most non-injurious falls have not been captured in your study (during follow-up).

We think we have been transparent in the methods that we have focused on gathering information on more serious falls within the trial, and have used the qualitative sample to test the feasibility of gathering information on falls not requiring ED attention or admission via diaries. However, we agree that this could be restated within the discussion and have added the following on page 27, line 463-466:

“Our findings suggest that accurately capturing prospective falls data may be challenging due to under-reporting, and yet, retrospective falls data collection does not fully reflect the incidence and impact of falls, particularly those which do not require an ED visit or hospital admission”

And:

“This study suggests that IDC may reduce the incidence of falls resulting in ED visits and hospital admissions in frail HD patients potentially by attenuating a decline in exercise capacity, physical activity behaviour and function at levels shown to be clinically meaningful in other long-term conditions”

On page 29, line 517-520.

It is also unclear how many participants experienced falls during the prospective follow-ups (i.e. how many single- and recurrent-fallers?).

Apologies for omitting this, within supplementary material 6, which relates to the falls outcomes, we have now added information on the number of non-fallers, fallers and frequent fallers as requested, and slightly reformatted the table so that it is displayed in the same way as the other supplementary outcomes. This table now appears as follows:

Supplementary material 6. Falls summary data and incidence of falls per person years.

	Usual care (n=27)	Exercise (n=24)
Number of Falls	11	5

Number (% of group) of non-fallers	19 (70)	20 (83)
Number (% of group) fallers (≥ 1 fall)	8 (30)	4 (17)
Number (% of group) frequent fallers (≥ 2 falls)	3 (11)	1 (4)
Person years	40.5	36
Incidence rate	0.27	0.14

Please clarify how the crude falls incidence rate ratio was calculated.

Thank you this is clarified as follows on page 11, lines 224-228:

“For falls, summary data, incident rate ratio (the ratio of the incidence rate in the exercise group divided by the incidence rate in the usual care group). and 95% confidence intervals were presented. Statistical analyses were performed using SPSS 24 (IBM UK Ltd, UK) and Stata 16 (StataCorp LCC,USA).”

Discussion, lines 448-450: I agree with your point that although falls diaries (completed by patients) are usually regarded as the gold standard for falls monitoring, in the context of CKD-5-HD, prospective data collection at the dialysis unit may be a more suitable approach. Perhaps you can take one step more and suggest that recording falls information routinely during the HD session (e.g. once a week) may be useful in clinical practice and would also allow researchers to rely on higher quality falls data (for both prospective and retrospective studies).

This is an important point and we have added the following text on page27, lines 466-472 in light of the reviewers comments, thank you.

“Given the high incidence of falls in this population, capturing falls data may be important in a future trial, and regular prospective recording of information relating to falls as a part of routine practice at the dialysis unit is recommended, in line with participant feedback.⁵ This would provide both clinicians and researchers with higher quality data for use in both prospective and retrospective studies, and to inform clinical care.”

We hope that we have adequately addressed each of the reviewers’ points and would like to thank them once again for taking the time to read and comment upon our manuscript.

VERSION 2 – REVIEW

REVIEWER	Adnan Sharif Queen Elizabeth Hospital Birmingham UK
REVIEW RETURNED	28-Jul-2020

GENERAL COMMENTS	Thank you - no further queries and questions.
---

REVIEWER	Danielle Kirkman Virginia Commonwealth University
REVIEW RETURNED	14-Aug-2020

GENERAL COMMENTS	The authors have adequately addressed the comments that I raised in my previous review and made the appropriate changes to their manuscript.
--

REVIEWER	Tobia Zanotto University of Illinois at Urbana-Champaign
REVIEW RETURNED	07-Aug-2020

GENERAL COMMENTS	The authors have satisfactorily addressed all of my comments/observations and the manuscript has been further improved. I believe the article is acceptable for publication in its current version.
---

VERSION 2 – AUTHOR RESPONSE

Many thanks to you and the reviewers for accepting our revised manuscript. We discuss the changes we have made in order to adhere to your editorial policies below and have also highlighted where changes have been made using tracked changes within an updated version of the manuscript.

"We are concerned that the data presented in the paper is not meeting BMJ standards of anonymisation. Tables 3-6 and figure 2 report exact ages of participants. Our latest guidelines consider exact age to be a direct identifier. Can you please ensure that the paper contains no direct identifiers and no more than two indirect identifiers?"

We have now removed descriptions of participants ages from Tables 3-6 and Figure 2 in order to comply with your editorial policies. We have also taken this opportunity to amend some minor typographical errors, which are marked in the track changes. The manuscript was at the upper limits of the word count allowance when originally submitted because it describes a mixed-methods study, and the full word count was required to adhere to the reporting guidance for both trials and qualitative studies. In responding to reviewers comments and editorial requirements we have now slightly exceeded this maximum word count. We hope this is acceptable, but would be very willing to make edits to the manuscript, with your guidance, should this be required.